# Enhancement of Magnetic and Tensile Mechanical Performances in Fe-Based Metallic Microwires Induced by Trace Ni-Doping

**DOI:** 10.3390/ma14133589

**Published:** 2021-06-27

**Authors:** Mingwei Zhang, Guanda Qu, Jingshun Liu, Mengyao Pang, Xufeng Wang, Rui Liu, Guanyu Cao, Guoxi Ma

**Affiliations:** 1School of Materials Science and Engineering, Inner Mongolia University of Technology, Hohhot 010051, China; 20191100205@imut.edu.cn (M.Z.); 20181800222@imut.edu.cn (G.Q.); 20181100183@imut.edu.cn (M.P.); 20191100209@imut.edu.cn (R.L.); 20181800242@imut.edu.cn (G.M.); 2Faculty of Materials and Manufacturing, Beijing University of Technology, Beijing 100022, China; Wangxf12333@emails.bjut.edu.cn; 3School of Materials Science and Engineering, Harbin Institute of Technology, Harbin 150001, China; guanyu_cao@stu.hit.edu.cn

**Keywords:** Fe-based metallic microwires, Ni-doping amount, nanocrystalline structure, magnetic property, tensile mechanical property

## Abstract

Herein, the effect of Ni-doping amount on microstructure, magnetic and mechanical properties of Fe-based metallic microwires was systematically investigated further to reveal the influence mechanism of Ni-doping on the microstructure and properties of metallic microwires. Experimental results indicate that the rotated-dipping Fe-based microwires structure is an amorphous and nanocrystalline biphasic structure; the wire surface is smooth, uniform and continuous, without obvious macro- and micro-defects that have favorable thermal stability; and moreover, the degree of wire structure order increases with an increase in Ni-doping amount. Meanwhile, FeSiBNi2 microwires possess the better softly magnetic properties than the other wires with different Ni-doping, and their main magnetic performance indexes of *M*_s_, *M*_r_, *H*_c_ and *μ*_m_ are 174.06 emu/g, 10.82 emu/g, 33.08 Oe and 0.43, respectively. Appropriate Ni-doping amount can effectively improve the tensile strength of Fe-based microwires, and the tensile strength of FeSiBNi3 microwires is the largest of all, reaching 2518 MPa. Weibull statistical analysis also indicates that the fracture reliability of FeSiBNi2 microwires is much better and its fracture threshold value *σ*_u_ is 1488 MPa. However, Fe-based microwires on macroscopic exhibit the brittle fracture feature, and the angle of sideview fracture *θ* decreases as Ni-doping amount increases, which also reveals the certain plasticity due to a certain amount of nanocrystalline in the microwires structure, also including a huge amount of shear bands in the sideview fracture and a few molten drops in the cross-section fracture. Therefore, Ni-doped Fe-based metallic microwires can be used as the functional integrated materials in practical engineering application as for their unique magnetic and mechanical performances.

## 1. Introduction

The atoms of amorphous alloys are arranged in a state of long-range disorder and short-range order, and the atoms are bonded by metallic bonds [1]. Under the similar composition condition, amorphous alloys have higher strength, elastic deformation capacity and relatively lower elastic modulus compared with crystalline alloys [2]. Meanwhile, as a kind of quasi-one-dimensional metallic materials, microscale metallic microwires can show excellent properties in the fields of mechanics and magnetism [3] and can be used as magnetic sensitive materials [4,5], magnetostrictive materials [6,7], magnetic refrigeration materials [8,9], electromagnetic shielding materials [10,11], lead frame materials [12,13], etc. However, due to the highly localized deformation of metallic microwires and the uneven nucleation in the shear band, the brittleness tendency appears at room temperature, which limits its engineering applications to some extent [14,15]. Therefore, it is essential to improve the mechanical properties of metallic microwires.

Compared with the traditional softly magnetic materials, Fe-based amorphous alloys possess obvious advantages due to their high-saturation magnetization, high permeability, low coercivity, low magnetic loss, and magnetic isotropy, etc., but the generally magnetic properties have yet to be enhanced [16]. Therefore, the researchers usually adopt the method of trace-elements doping and modulation processing state in order to improve softly magnetic properties of the metallic microwires. Wang A. D. et al. [17] prepared Fe-P-C-B-Si amorphous alloys system with high *M_s_*. by adjusting the contents of B, Si and P, and the *M_s_* of the amorphous alloys system was improved, which could reach 1.68 T. Shi Z. G. et al. [18] researched Fe-Co-B-P-C alloys, found that the introduction of Co element causes ferromagnetic coupling between Co and Fe, and that the saturation magnetization of the amorphous alloys system increases from 1.56 to 1.79 T. Zhang Y. et al. [19] researched the amorphous (Fe_0.9_Co_0.1_)_72.7_Al_0.8_Si_13.5_Cu_1_Nb_3_B_8_V_1_ alloys, where the annealing temperature increases, the thickness of the amorphous layer gradually decreases and is advantageous to the exchange coupling effect between adjacent grain, and the softly magnetic properties of the amorphous alloys have been improved. Wang C. X. et al. [20] found that after magnetic field annealing, coercivity of Fe_80_Si_9_B_11_ alloys decreased by 71.5%, and hysteresis loss decreased by 68.2%, further improving its softly magnetic properties. Presently, there are few studies on the effect of trace-element doping on the magnetic properties of Fe-based metallic microwires, and the influence mechanism is still unclear.

Presently, the previous researchers have prepared Fe-based, Co-based, Cu-based and Gd-based metallic microwires, which possess potential application in military defense, aerospace, automotive electronics and other fields. Compared with bulk amorphous alloys, metallic microwires usually adopt trace-elements doping to regulate its structure to improve its mechanical properties [21]. Zhang Y. et al. [22] reported that Pr element doping improved the mechanical properties of Cu-Zr-Ti metallic microwires, and the tensile strength and elongation of the microwires reached 2.07 GPa and 0.92%, respectively. Chirica H. et al. [23] found that appropriate annealing temperature can effectively improve the mechanical properties of metallic microwires. Additionally, phase separation and surface strengthening can also have a certain role on the improvement of its mechanical properties [24,25]. At the same time, the research on improving the mechanical properties of Fe-based metallic microwires by trace-elements doping is also not systematic and comprehensive, and the fracture deformation mechanism based on the metallic microwires fracture morphology analysis is still unclear, so it is very necessary to carry out the above research work.

In this paper, we aim to study the effect of Ni-doping amount on microstructure, magnetic and tensile mechanical properties of Fe-based metallic microwires. The influences of structural order degree, magnetic property index and mechanical property index of microwires before and after Ni-doping were significantly compared and enhanced. Based on the analysis of fracture reliability and fracture morphology, the fracture deformation mechanism of microwires was accordingly illustrated. Finally, it also can provide the reliable technical support for the practical engineering application of Fe-based metallic microwires.

## 2. Materials and Methods

The master alloys were proportioned according to the nominal composition Fe_78-x_Si_13_B_9_Ni_x_ (*x* = 0, 1, 2, 3, in at.%), and the purity of the raw materials is above 99.999%. In this paper, the symbols of FeSiB, FeSiBNi1, FeSiBNi2 and FeSiBNi3 are used to represent the Fe-Si-B-Ni series metallic microwires. Firstly, the master alloys were smelted by magnetron-tungsten-electrode vacuum arc-melting furnace. Before smelting, the furnace was vacuumed to 10^−3^ Pa, and the protective gas of high-purity Ar, by electromagnetic stirring and remelting 5–6 times, preforms rods of master alloys with diameter of Φ8 mm, and a length of around 100 mm was obtained by copper mold suction casting method. Subsequently, the precision rotated-dipping process was firstly conducted by an induction coil to heat the top of the master alloys rods above the melting point and adjust the relevant process parameters. The speed of the Cu wheel was 1700 r/min, the feed speed of the molten pool was 25 μm/s, and the heating power of the power supply was 20 kW. The Cu wheel wedge flange was used to dip quantitative melt in the molten pool, and the micro-scale metallic microwires with the length of about 60 cm and the diameter of Φ35–60 μm were formed after being thrown out by inertia.

The phase composition analysis of the microwires was carried out with the Rigaku D/MAX-2500 X-ray diffraction (XRD) analyzer with an accelerating voltage of 40 kV, the scanning angle (2*θ*) range of 20°–100°, the maximum power of 18 kW and the scanning velocity of 3°/min, and the X-ray machine uses high-vacuum X-ray ceramic tube sealed by cathode emission electron filament and rotating anode copper target under high voltage. The thermal stability analysis of Fe-based metallic microwires was performed on SETARAM DSC131 EVO type differential scanning calorimeter (DSC), with the heating rate and cooling rate being 10 and 20 °C/min, respectively and the temperature range from room temperature being at 1000 °C. The surface and fracture morphology based on the secondary electron imaging analysis of Fe-based metallic microwires was conducted on FEI QUANTA 650 FEG type scanning electron microscope (SEM) combining energy disperse spectroscopy (EDS), with accelerating voltage of 20 kV. The transmission samples were prepared with the Leica EM RES102 multifunctional ion thinner, and the JEOL JEM2010 transmission electron microscope was used to analyze the micromorphology, high-resolution images and selected-area electron diffraction patterns of the metallic microwires. The magnetic properties of cluster metallic microwires were measured on Quantum Design SQUID-VSM-type magnetic performance measurement system (MPMS), with the maximum external magnetic field of 5 T. Moreover, the tensile properties of metallic microwires were conducted by Instron 5943 type electronic universal tensile testing device, and the wires were fixed with self-designed fixtures. Due to the fine and brittleness of the wires, tensile samples of the wires were prepared according to ASTM D3379-75. The extension rate is 0.2 mm/min. Simultaneously, Weibull statistical methods were used for analyzing the tensile property and fracture reliability.

## 3. Results

### 3.1. Microstructure and Magnetic Properties of Fe-Based Metallic Microwires

Figure 1 shows the XRD patterns of Ni-doped Fe-based metallic microwires. It can be seen that the sharp diffraction peaks appear around the angle of 2*θ* of 45°, 66° and 83°, respectively. Compared with the PDF standard card, a large number of α-Fe phase and a small number of Fe_3_Si phase exist in the Fe-based metallic microwires, indicating that the as-prepared Fe-based metallic microwires have amorphous and nanocrystalline biphasic structure characteristics.

Figure 2 exhibits DSC curves in heating and cooling stages of Ni-doped Fe-based metallic microwires. With the increase in temperature, two exothermic peaks appear in the DSC curves of Ni-doped Fe-based metallic microwires, indicating that the secondary crystallization phenomenon occurs in the heating stage. Thermophysical parameters such as glass transition temperature *T*_g_, primary crystallization temperature *T*_x1_, secondary crystallization temperature *T*_x2_ and mixing enthalpy -Δ*H* can be obtained from the curves, as shown in Table 1.

The mixing enthalpy −Δ*H* of Ni-doped Fe-based metallic microwires increases first and then decreases, and the −Δ*H* of FeSiBNi2 is the largest, which is 59.30 J/g, indicating FeSiBNi2 metallic microwires require higher energy from metastable status to low-energy metastable status, and the atomic relaxation phenomenon is more obvious. In addition, the glass-transition temperature region (namely Δ*T* = *T*_X1_ − *T*_g_) can quantitatively characterize the amorphous formation ability of the microwires. The Δ*T* of FeSiB metallic microwires is 66.30 °C, and the Δ*T* of FeSiB metallic microwires is slightly reduced after Ni-doping. The Δ*T* values of Ni-doped Fe-based metallic microwires are all larger than 55 °C; therefore, these wires exhibit the excellent thermal stability.

Figure 3 illustrates the SEM surface morphology and EDS energy spectrum of Ni-doped Fe-based metallic microwires. Fe-based metallic microwires are uniform, smooth, without obvious Rayleigh wave defects, with a diameter of about 45 μm, as shown in Figure 3a,c,e,g. According to EDS energy spectrum analysis, the energy spectrum peaks of each element are located accurately, as shown in Figure 3b,d,f,h. Therefore, the rotated-dipping Fe-based metallic microwires exhibit a relatively superior roundness without the chemical segregation by the coaction of surface tension and melt gravity.

Figure 4 illustrates the transmission electron microscopy (TEM) morphology, selected-area electron diffraction pattern (SAED), HRTEM morphology, Fourier-inverse Fourier (FFT-IFFT) transformation, ACF autocorrelation diagrams, #1 region and #2 region HRTEM morphology of FeSiB microwires and statistics diagram of the interplanar spacing and atomic size. TEM microstructure characterization of FeSiB metallic microwires appears as distinct black regions. The reason for this contrast is the composition segregation in materials with the strong ability of scattering incident electrons, as shown in Figure 4a. The SAED possesses the characteristics of polycrystalline ring and diffraction spot, which proves the matrix is an amorphous and existing nanocrystalline structure, as shown in Figure 4b. The HRTEM morphology shows that there exist obvious crystallization phenomena in #1 region and #2 region, and the FFT transformation also has obvious polycrystalline diffraction spots.

The autocorrelation function (ACF) is used to evaluate the degree of structure order in HRTEM, which can be expressed by Equation (1):(1)$$\Psi =\frac{\varsigma }{\kappa }\times 100\%$$
where *ς* is the number of pictures in order and *κ* is the number of HRTEM divisions.

In the ACF image, there are fine stripes in the red region, which is an ordered region, and the blue region has no obvious characteristics, which is a disordered region. According to the order degree statistics, the structure order *ψ* of the FeSiB microwires is 12.5%. Due to the regional crystallization of FeSiB microwires, the nanocrystals in FeSiB microwires were characterized. Figure 4d,g are the partial enlargement of Figure 4c #1 region and #2 region and calculate the interplanar spacing and atomic size in these two regions. In the FeSiB HRTEM image, the interplanar spacings of #1 region and #2 region are 0.193 and 0.213 nm, and comparison of the PDF standard card shows that the two regions correspond to the Fe_3_Si phases and α-Fe phases, respectively, and the internal atomic radii are 0.115 nm and 0.130 nm, respectively.

Figure 5 indicates the TEM morphology, selected-area electron diffraction pattern, HRTEM morphology, Fourier-inverse Fourier (FFT-IFFT) transformation, ACF autocorrelation function diagrams, #3 region and #4 region HRTEM morphology of FeSiBNi2 microwires and statistics diagram of the interplanar spacing and atomic size. It can be seen from the figure that the “lamellar fish scale” appears in the TEM morphology of FeSiBNi2 microwires. The reason for the appearance of this morphology is that the nanocrystals increase the local hardness; therefore, in the process of ion thinning, the atoms in the region of greater hardness are difficult to be removed and present the mass-thickness contrast. Meanwhile, polycrystalline rings and diffraction spots appeared in SAED, indicating the matrix is amorphous and the presence of a large number of nanocrystals. There is a large number of striped organizations in the HRTEM image of FeSiBNi2, both polycrystalline rings and diffraction spots appear in the FFT image, and an ordered striped organization also appears in the IFFT image, as shown in Figure 5c. The degree of order is evaluated quantitatively through ACF image, and after calculation, the degree of order *ψ* is 71.9%. Figure 5d,g are the partial magnifications of #3 region and #4 region in Figure 5c which calculate the interplanar spacing and atomic size. In the FeSiBNi2 HRTEM image, the interplanar spacings of #3 region and #4 region are 0.203 and 0.197 nm, and comparison of the PDF standard card shows that the two regions correspond to the α-Fe phases and Fe_3_Si phases, respectively, and the internal atomic radii are 0.104 and 0.107 nm, respectively. Therefore, it is a typical amorphous and nanocrystalline biphasic structure of Fe-based microwires, and with increase in Ni-doping content, the number of internal nanocrystals increases.

Figure 6 exhibits the curves of generally magnetic properties of Ni-doped Fe-based metallic microwires. The saturation magnetization *M*_s_ of FeSiB is the highest, which is 174.06 emu/g. After doping with Ni element, the *M*_s_ of Fe-based metallic microwires decreases somewhat, and the *M*_s_ of FeSiBNi3 is the smallest, which is 165.81 emu/g, as shown in Figure 6a. The residual magnetization *M*_r_, coercivity *H*_c_, and permeability *μ*_m_ of FeSiB are the highest, which are 10.82 emu/g, 33.08 Oe and 0.43, respectively. Ni-doping causes the *M*_r_, *H*_c_ and *μ*_m_ of Fe-based metallic microwires to decrease slightly, as shown in Figure 6b. In addition, the generally magnetic index parameters of Ni-doped Fe-based metallic microwires were statistically analyzed, as shown in Table 2. The Ni atoms partially replace the Fe atoms in the alloys, and the *M*_s_ of the metallic microwires in the alloy system is all above 165 emu/g, but it decreases slightly. This is due to the negative binding energy of Ni element, which tends to form a Fe vacancy in the alloy system after addition, resulting in a local structural distortion that reduces the saturated flux density of FeSiB alloys, reduces the local magnetic moment and reduces *M*_s_ [26,27]. On the basis of inherent composition, due to the presence of nanocrystals in the material, nanocrystals distribute the magnetocrystalline anisotropy of the material, resulting in a smaller *H*_c_ and better softly magnetic properties [28]. In conclusion, FeSiBNi2 has the best softly magnetic properties because it has the smaller *H*_c_ and *M*_r_ and the larger *M*_s_ and *μ*_m_.

### 3.2. Tensile Properties and Fracture Morphology of Fe-Based Metallic Microwires

Figure 7 shows typical tensile stress–strain curves of Ni-doped Fe-based metallic microwires. From the figure, the tensile strengths of FeSiB, FeSiBNi1, FeSiBNi2 and FeSiBNi3 are 1247, 2050, 2462 and 2518 MPa, respectively. At the same time, comparing the tensile curves, it was found that the Fe-based microwires before and after Ni-doping curves exhibit good elastic deformation characteristics at the initial stage of tensile deformation. As the applied load increases, the tensile stress–strain curves have a certain offset, which eventually leads to fracture failure [29]. Therefore, Ni-doped Fe-based metallic microwires exhibit excellent tensile mechanical properties and exhibit brittle characteristics.

Figure 8a,c,e,g illustrates the sideview fracture morphology of Ni-doped Fe-based metallic microwires. As can be seen, the fracture of Ni-doped Fe-based metallic microwires is an oblique fracture, and there are a large number of shear bands that were found on the sideview fracture. The formation of shear bands occurs at the beginning of tensile deformation, and microwires undergo highly localized shear deformation. These deformations are attributed to creation and annihilation of free volume [30] and thermal-softening [31]; therefore, volume expansion and structural disorder occur at atomic scale [32]. In a local region, the atoms rearrange themselves and form extremely fine and highly localized shear bands. Once formed, these highly localized shear bands expand rapidly until the microwires fracture. The sideview fracture morphology of the microwires shows that the fracture angle decreases as the Ni-doping amount increases. The fracture angle *θ* of FeSiB is the largest, which is 52°; the fracture angle *θ* of FeSiBNi3 is the smallest, which is 42°. This is due to the increase in Ni-doping making the nanocrystals content increase, leading to the critical normal fracture stress on the shear plane being much smaller than the shear stress and making the fracture angle *θ* tend to decrease [33].

Indeed, the partial stepped morphology similar to the flat fracture appears on the sideview fracture of FeSiB metallic microwires, which can be attributed to the uneven distribution of nanocrystals in the material and the weak binding force between nanocrystals and the matrix material, leading to crack propagation at the nanocrystals or material defects and eventually a fracture, as shown in Figure 8a. Figure 8b,d,f,h is the cross-section fracture morphology of Ni-doped Fe-based metallic microwires. According to these figures, the cross-section fracture morphology is composed of two parts: crack-extension region and shear-deformation region. Due to the massive release of residual stress in the metallic microwires at the moment of tensile fracture, the temperature in part of the main shear region increases, and the surface viscosity decreases; therefore, the vein-shaped pattern appears in the crack extension zone. The temperature in the fracture deformation region rises instantaneously, causing liquid metal splashing when fracture occurs, meaning a small amount of metal drops is formed at the fracture after cooling. From the fracture characteristics of Fe-based metallic microwires emerges a typical vein-shaped pattern in the crack extension region, which is caused by the material bearing a great deal of strain energy [34]. At the same time, the presence of a certain amount of nanocrystalline in the microwires can improve the plasticity of the alloys [35].

Spaepen F. [36] explained the rheological behavior in amorphous alloys with free volume theory; that is, in the “free volume model” deformation mechanism of amorphous alloys, the degree of disorder of the system is measured by the free volume. Therefore, the tensile deformation process of Ni-doped Fe-based metallic microwires can be divided into four stages, corresponding to Figure 9a–d. (1) Elastic deformation is manifested in the initial stage of Fe-based metallic microwires fracture. (2) Under the uniaxial tension tensile stress, the free volume gathered and the rheological defect fusion cause a partial rearrangement of atoms to form nanoscale micropores. (3) As the tensile stress continues to increase, the micropores continue to proliferate and grow, and local shear fracture occurs between the micropores. At the same time, the free volume in the system makes the alloy partly viscous flow, the energy provided by the external force in the rheology is transformed into internal energy, and a vein-like pattern is formed when the alloy is cooled, as shown in Figure 9e–h. (4) When the tensile stress increases to the limit of elastic deformation, the shear band rapidly expands under the action of the stress and eventually breaks.

The Weibull statistical method can reflect the distribution and uniformity of strength values of brittle materials and characterize the distribution of defects inside the materials and the sensitivity to defects. Therefore, the Weibull statistical method is used to quantitatively analyze the fracture reliability of Ni-doped Fe-based metallic microwires [37], which can be expressed as:(2)$$P_{f}=1-\exp\left[-\int^{\;}v{\left(\frac{\sigma -\sigma_{u}}{\sigma_{0}}\right)}^{m}\right]dv$$
where *P*_f_ is the probability of fracture under a certain stress; *σ*_0_ is the characteristic stresses associated with the material; *σ*_u_ is the fracture threshold value; *v* is the sample volume; and *m* is the Weibull modulus. The threshold value *σ*_u_ and modulus *m*, which represent the tensile strength distribution, can be obtained by Weibull statistics. The larger value of *σ*_u_ indicates that the material almost does not fracture under this stress and possesses higher stability. The larger value of *m*, which is the fracture strength of material, is distributed in a narrow range and exhibits higher fracture reliability.

The Weibull statistical function is linearized, when *σ*_u_ = 0 or ≠ 0, and the treated statistical function corresponds to the two- and three-parameter Weibull statistical function, respectively, and the expression is [38]:(3)$$\ln\left\{\ln\frac{1}{\left(1-p_{f}\right)}\right\}=m\ln\left(\sigma -\sigma_{u}\right)-m\ln\sigma_{0}$$

The brittle fracture wires also conform to lognormal distribution, and the lognormal model can also be used to describe their fracture reliability. The strength–stress model of lognormal distribution can be expressed by the following equation [39]:(4)$$P_{f}=\frac{1}{2}\left[1+\mathrm{erf}\left(\frac{\ln\left(\sigma \right)-k}{s\;\sqrt{2}}\right)\right]$$
where *k* is the mean value of the data and *s* is the standard deviation coefficient. Among these, the standard deviation coefficient *s* is the degree of experimental data dispersion, and the smaller value of *s* indicates that the less experimental data deviates from average value.

Figure 10 reveals the two- and three-parameter fitting and logarithmic fitting curves of Fe-based metallic microwires based on Weibull statistics. According to Figure 10a,c,e,g, the two- and three-parameter fracture modulus *m* of FeSiB is 11.11 and 8.47, respectively. The two- and three-parameter fracture modulus decreases significantly after Ni-doping, indicating that the fracture strength distribution range of Ni-doped Fe-based metallic microwires becomes wider and the fracture reliability decreases. The fracture threshold value *σ*_u_ of FeSiB is 267 MPa, while the *σ*_u_ of Fe-based microwires is significantly increased after Ni-doping. The results show that the Ni-doped Fe-based microwires have better stability, and the *σ*_u_ of FeSiBNi2 is up to 1488 MPa. The fitting results based on normal distribution show that the *s* of FeSiB is 0.07378, which is less than *s* of Fe-based metallic microwires after Ni-doping, indicating that Ni-doping increases the degree of dispersion and the distribution range is wider. Meanwhile, the mean value *k* of Ni-doped Fe-based metallic microwires increases significantly, indicating that the overall tensile strength of metallic microwires becomes better after Ni-doped. The mean value *k* of FeSiBNi3 is the largest, reaching 7.84469, as shown in Figure 10b,d,f,h. The relevant Weibull parameters obtained from Figure 10 are listed in Table 3. Therefore, Ni-doping significantly increases the tensile strength of Fe-based metallic microwires, which effectively improves safety and stability of the metallic microwires.

Moreover, due to Ni-doping prompted in the amorphous matrix forming a more orderly nanophase, nanophase does not have dislocations and other defects; it has a nearly perfect crystal structure; nanophases are distributed in the whole metallic wires; it is beneficial to restrain the formation and expansion of shear bands in the tensile process; and this acts as diffusion and reinforcement and significantly improves the tensile strength of the metallic wires. Meanwhile, the more the number of nanocrystals, the higher the viscosity in the plastic deformation zone, and the higher the nonuniform shear rheological resistance; therefore, the tensile strength is improved.

## 4. Conclusions

Conclusively, we investigated the microstructure, generally magnetic properties and tensile mechanical behavior of Fe-based metallic microwires before and after Ni-doping, and the following conclusions can be drawn:

(1) The structure of rotated-dipping Ni-doped Fe-based microwires is amorphous and nanocrystalline biphasic structure, exhibiting the excellent thermal stability. The surface of Fe-based microwires is smooth, uniform, continuous and without obvious defects. Moreover, Ni-doped Fe-based microwires possess excellent softly magnetic properties and the magnetic performance indexes of *M*_s_, *M*_r_, *H*_c_ and *μ*_m_ of FeSiBNi2 microwires are 174.06 emu/g, 10.82 emu/g, 33.08 Oe and 0.43, respectively.

(2) The tensile strength of Fe-based microwires increases with the increase in Ni-doping amount and exhibits the largest tensile strength of 2518 MPa: namely, the increase in tensile strength is probably related to the degree of structure order. Meanwhile, FeSiBNi2 microwires have better fracture stability, and their fracture threshold value *σ*_u_ is 1488 MPa.

(3) Ni-doped Fe-based metallic microwires show the brittle fracture on a macroscopic level, and the angle of sideview fracture *θ* decreases with Ni-doping amount increasing, which also reveals the certain plasticity due to a certain amount of nanocrystalline in the microwires structure. Meanwhile, there is a huge amount of shear bands in the sideview fracture and a few molten drops in the cross-section fracture. During tensile deformation process, the free volume aggregation and rheological defect fusion result in the formation of fine micropore. Subsequently, the propagation and expansion of micropores promote that the microcracks propagate rapidly, and a fracture occurs under shear stress.

(4) Ni-doped Fe-based metallic microwires possess excellent softly magnetic and tensile mechanical performances, and it can be concluded that the Ni-doped type microwires have the potential engineering application value in different fields, including magnetic sensitive detection, magnetostriction, electromagnetic measurement, lead frame, etc.

## Figures and Tables

**Figure 1 materials-14-03589-f001:**
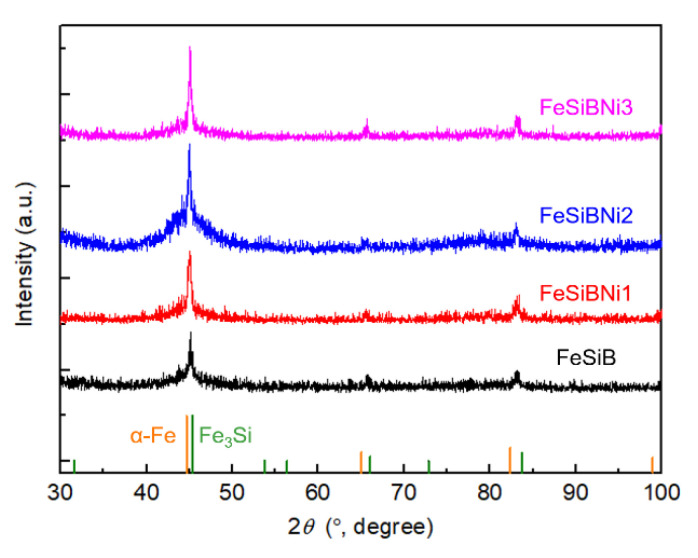
X-ray diffraction (XRD) patterns of Ni-doped Fe-based metallic microwires.

**Figure 2 materials-14-03589-f002:**
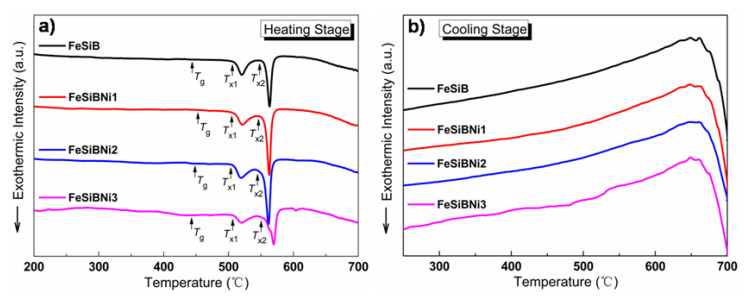
Differential scanning calorimetry (DSC) curves of Ni-doped Fe-based metallic microwires: (**a**) heating stage and (**b**) cooling stage.

**Figure 3 materials-14-03589-f003:**
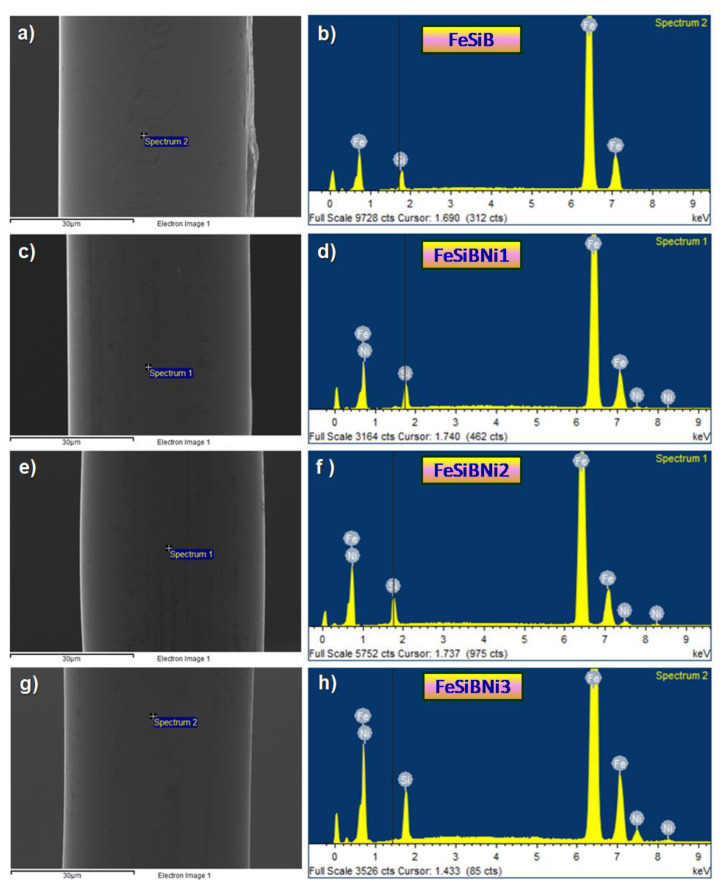
Scanning electron microscope (SEM) surface morphology and energy disperse spectroscopy (EDS) analysis of Ni-doped Fe-based metallic microwires: (**a**,**b**) SEM and EDS of FeSiB; (**c**,**d**) SEM and EDS of FeSiBNi1; (**e**,**f**) SEM and EDS of FeSiBNi2; and (**g**,**h**) SEM and EDS of FeSiBNi3.

**Figure 4 materials-14-03589-f004:**
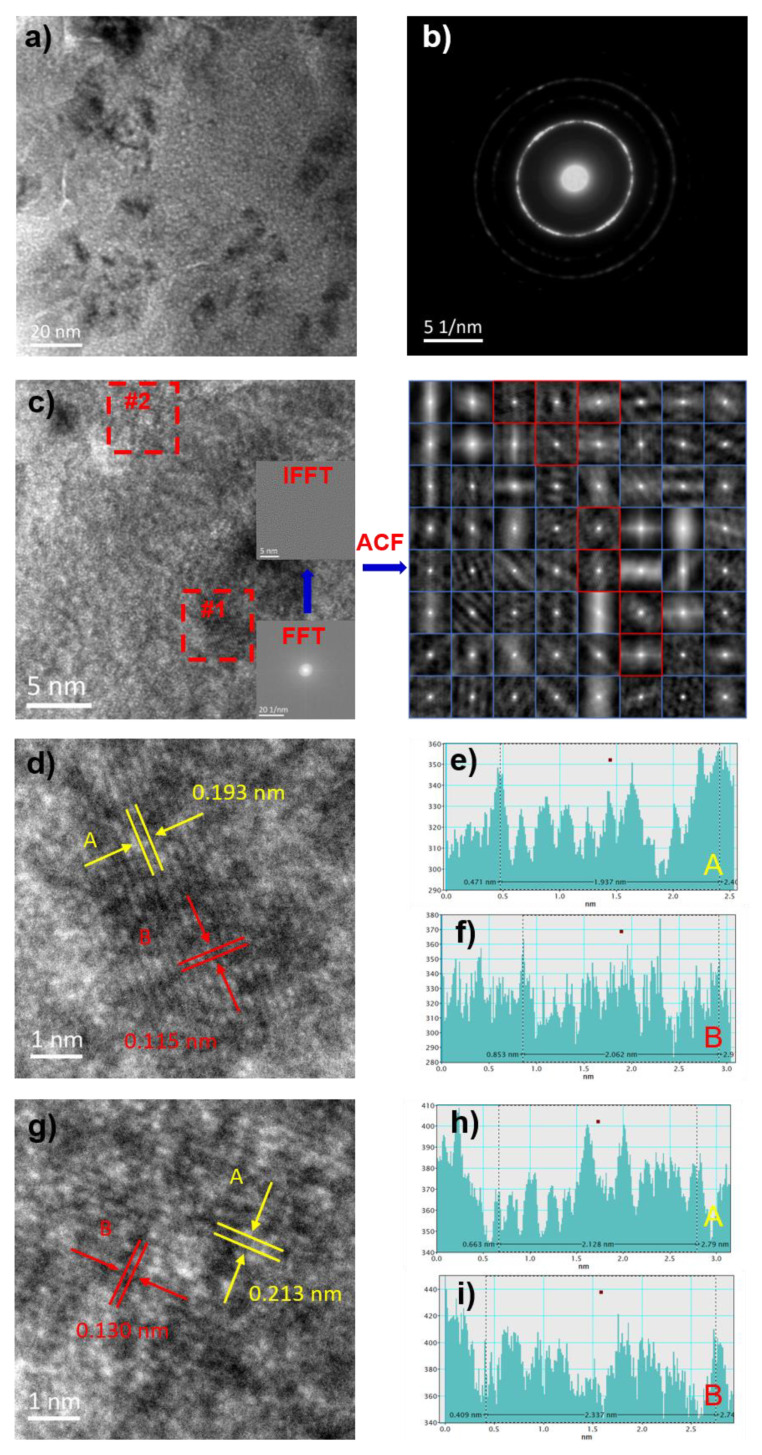
Transmission electron microscopy (TEM) microstructural characterization of FeSiB metallic microwires, HRTEM images and statistics diagram of the interplanar spacing and atomic size corresponding to the nanocrystalline: (**a**) TEM morphology; (**b**) SAED; (**c**) HRTEM images, FFT, IFFT and ACF; (**d**) HRTEM images of #1 region; (**e**,**f**) statistics diagram of the interplanar spacing and atomic size corresponding to the nanocrystalline in #1 region; (**g**) HRTEM images of #2 region; and (**h**,**i**) statistics diagram of the interplanar spacing and atomic size corresponding to the nanocrystalline in #2 region.

**Figure 5 materials-14-03589-f005:**
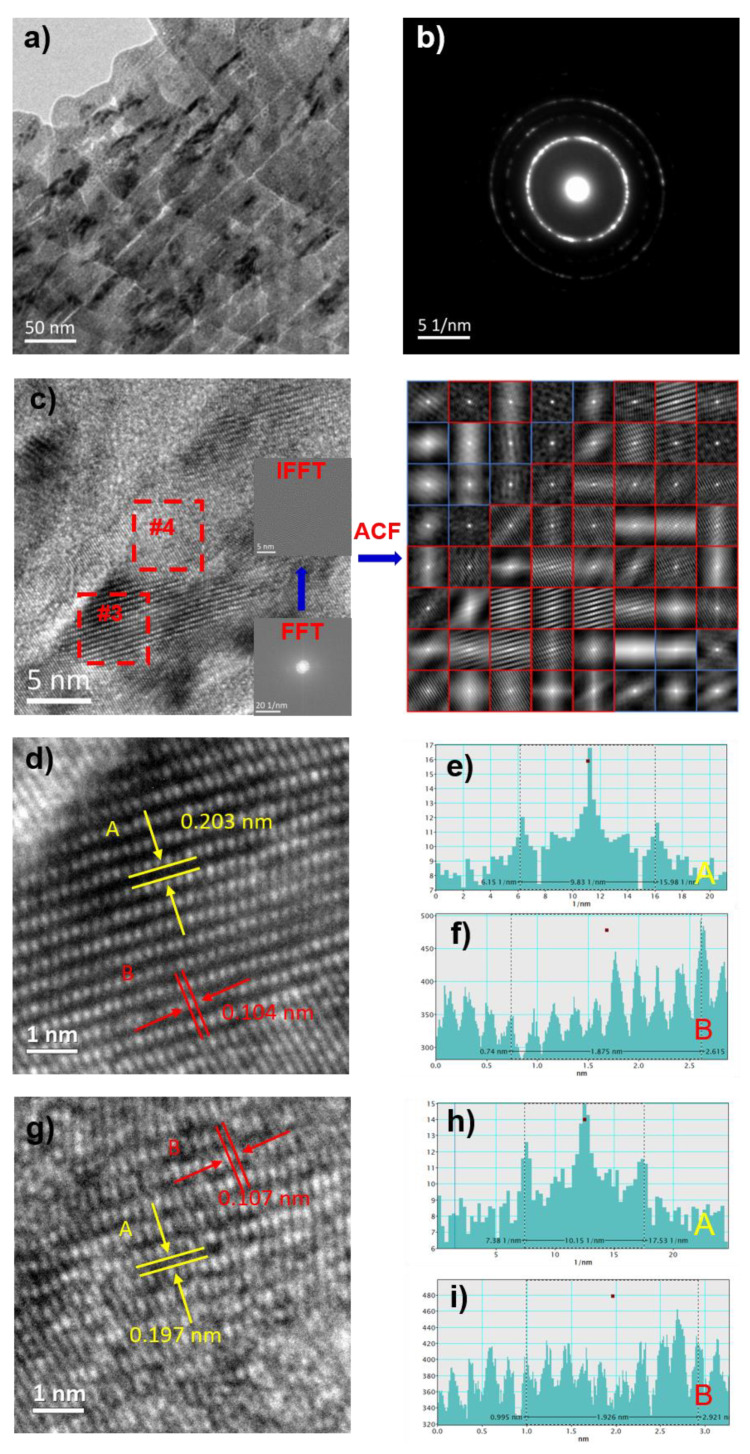
TEM microstructural characterization of FeSiBNi2 metallic microwires, HRTEM images and statistics diagram of the interplanar spacing and atomic size corresponding to the nanocrystalline: (**a**) TEM morphology; (**b**) SAED; (**c**) HRTEM images, FFT, IFFT and ACF; (**d**) HRTEM images of #3 region; (**e**,**f**) statistics diagram of the interplanar spacing and atomic size corresponding to the nanocrystalline in #3 region; (**g**) HRTEM images of #4 region; and (**h**,**i**) statistics diagram of the interplanar spacing and atomic size corresponding to the nanocrystalline in #4 region.

**Figure 6 materials-14-03589-f006:**
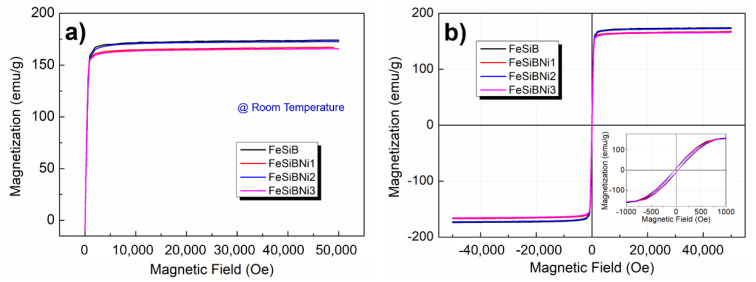
Comparison of the generally magnetic properties of Ni-doped Fe-based metallic microwires: (**a**) general magnetization curves and (**b**) hysteresis loops.

**Figure 7 materials-14-03589-f007:**
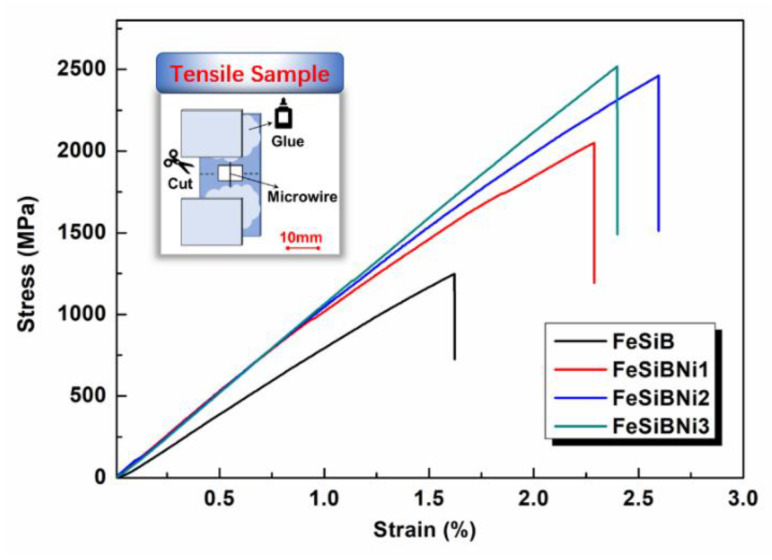
Typical tensile stress–strain curves of Ni-doped Fe-based metallic microwires. The inset is the tensile sample appearance.

**Figure 8 materials-14-03589-f008:**
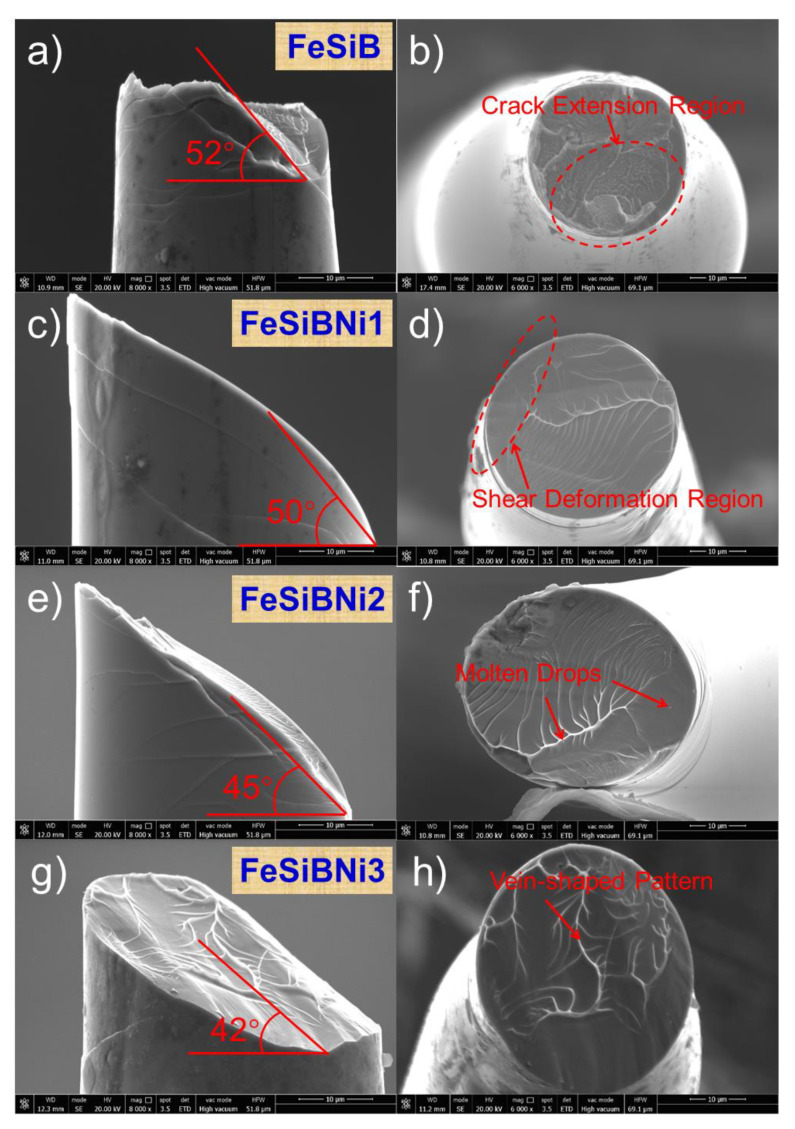
Sideview and cross-section fracture morphology of Ni-doped Fe-based metallic microwires: (**a**,**b**) FeSiB; (**c**,**d**) FeSiBNi1; (**e**,**f**) FeSiBNi2; and (**g**,**h**) FeSiBNi3.

**Figure 9 materials-14-03589-f009:**
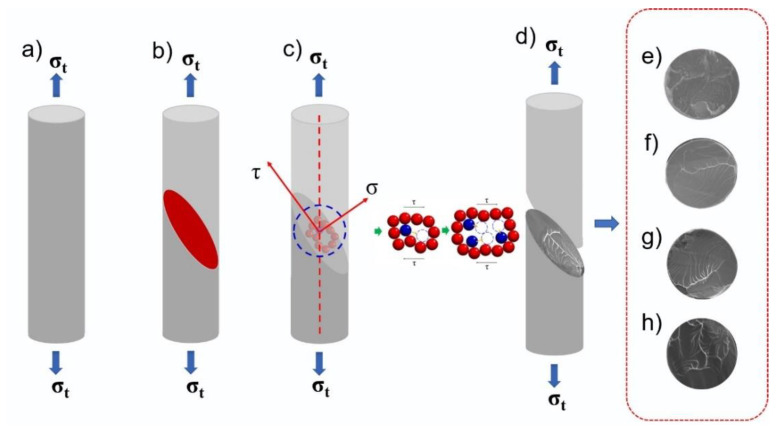
Schematic diagram of tensile fracture process of Fe-based metallic microwires: (**a**–**d**) are schematic diagrams of different fracture stages, and (**e**–**h**) are actual fracture patterns, respectively.

**Figure 10 materials-14-03589-f010:**
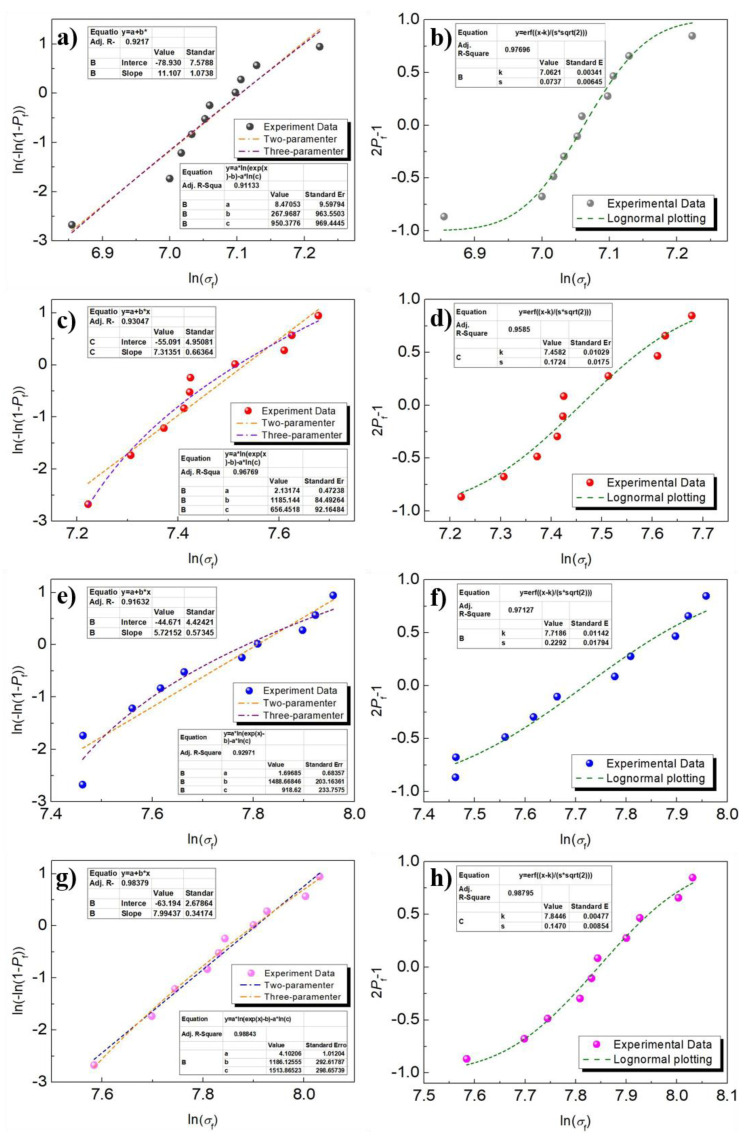
Fracture reliability analysis obtained by multitype plotting of Ni-doped Fe-based metallic microwires: (**a**,**b**) FeSiB, two- and three-parameter Weibull statistics and lognormal plotting; (**c**,**d**) FeSiBNi1, two- and three-parameter Weibull statistics and lognormal plotting; (**e**,**f**) FeSiBNi2, two- and three-parameter Weibull statistics and lognormal plotting; and (**g**,**h**) FeSiBNi3, two- and three-parameter Weibull statistics and lognormal plotting.

**Table 1 materials-14-03589-t001:** Statistical comparison of the thermophysical parameters of Ni-doped Fe-based metallic microwires.

Abbreviations of Wire Composition	*T*_g_ (°C)	*T*_x1_ (°C)	*T*_x2_ (°C)	Δ*T* (°C)	−Δ*H* (J/g)
FeSiB	438.41	504.71	541.32	66.30	51.30
FeSiBNi1	447.46	502.92	551.38	55.46	56.11
FeSiBNi2	443.29	506.27	546.54	62.98	59.30
FeSiBNi3	441.81	503.94	538.16	62.13	43.14

**Table 2 materials-14-03589-t002:** Statistics of the generally magnetic parameters of Ni-doped Fe-based metallic microwires.

Abbreviations of Wire Composition	*M*_s_ (emu/g)	*M*_r_ (emu/g)	*H*_c_ (Oe)	*μ* _m_
FeSiB	174.06	10.82	33.08	0.43
FeSiBNi1	167.04	10.25	32.77	0.32
FeSiBNi2	172.70	9.68	31.97	0.40
FeSiBNi3	165.81	10.79	32.47	0.33

**Table 3 materials-14-03589-t003:** Fracture reliability parameters statistics of Ni-doped Fe-based metallic microwires.

Fitting Type		FeSiB	FeSiBNi1	FeSiBNi2	FeSiBNi3
Weibull statistics	two-parameter	*m* = 11.11	*m* = 7.31	*m* = 5.72	*m* = 7.99
	three-parameter	*m* = 8.47	*m* = 2.13	*m* = 1.69	*m* = 4.10
		*σ*_u_ = 267 MPa	*σ*_u_ = 1185 MPa	*σ*_u_ = 1488 MPa	*σ*_u_ = 1186 MPa
Lognormal plotting		*k* = 7.06215	*k* = 7.45825	*k* = 7.71868	*k* = 7.84469
		*s* = 0.07378	*s* = 0.17246	*s* = 0.22924	*s* = 0.14708

## Data Availability

The data that support the findings of this study are available from the corresponding author, Jingshun Liu, upon reasonable request.

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
