# Peer review of "Enhancement of Magnetic and Tensile Mechanical Performances in Fe-Based Metallic Microwires Induced by Trace Ni-Doping"

_materials, 2021, doi:10.3390/ma14133589_

Round 1

Reviewer 1 Report

The manuscript deals with experimental investigations and analyses on the microstructure and magnetic and mechanical properties of iron-based microwires with different doping by nickel. The results described are of interest and of practical use. However, the following improvements are recommended before publication to raise the value of the manuscript:

(1) The radiation (Cu, Mo, etc.) used for the XRD analysis should be reported in the material and method section.

(2) In the description of Figure 1 the authors mention that sharp diffraction peaks are visible. The Miller index related to each of these peaks should be reported in the figure instead of the used symbols. To distinguish the phases, the related PDF card should be also reported in the figure and mentioned in the text as well as in the reference section with the related number.

(3) The strategy for fabricating the wires should be presented more clearly or an appropriate reference to a source should be made.

(4) The authors mention that” Compared with the standard PDF card, a large number of α-Fe phase and a small number of Fe3Si phase exist in the Fe-based metallic microwires”. This should be better explained. Moreover, the authors say that this indicates the presence of amorphous and nanocrystalline biphase structure. This statement should be checked. In fact, the presence of the nanocrystalline phase is indicated by the sharp diffraction peaks, while the presence of the amorphous phase is indicated by the broad halo around 45° 2 theta angle, visible in the FeSiBNi2 and FeSiBNi1 samples, where it is superimposed to the first sharp diffraction peak. The authors should better analyse and discuss the XRD results, with particular attention to the presence of the amorphous and nanocristalline phases, correlating them to the TEM results (SAED).

(5) The presentation of images 4, 5, 6, 7 and in particular 10 should be improved as some of the text is not readable and colours are not clear (e.g. the colour “blue”).

(6) All symbols and abbreviations used in the text and equations (e.g. "M_s", “V”, “v”) and in figures (e.g. "M" and "H" in Figure 6b) should be briefly explained.

(7) How many samples were used for the individual experiments?

(8) For "phi" in equation (1), "psi" is used in text. This should be consistent.

(9) Equation (2) could be omitted as it is not used or discussed further

(10) The term "side fracture" is not clear and should be replaced by a more appropriate term.

(11) The derivation of the Weibull distribution (line 312 ff.) used should be presented more comprehensibly. In the present form and with the sources given, this is not comprehensible and the development of equation (5) is unclear.

(12) The results in Figure 10 should be discussed more in detail. The authors leave it to the reader to interpret the diagrams.

(13) The text contains some very long sentences and some sentences that are unclear due to incorrect grammar. This should be improved.

Author Response

Thanks for the reviewer’s comment. According to the suggestion of reviewer, I have changed relevant content in revised MS. Please see the attachment.

Reviewer 2 Report

Line 18, page 1

What does the term "the degree of wire structure order" mean?

Line 24, .... page 1

Normally, tensile strength values ​​are written rounded to an integer (whole numbers).

Line 99, .... page 3

  • Explain in detail the technological process of making microwires.
  • I don’t think this technology is generally known to most readers of the journal.
  • Also indicate the dimensions of the resulting microwires.
  • Since the microstructure of the microwires depends on the technological parameters of the process, these must also be given.

Line 119, .... page 3

  • How the tensile test was performed, especially the clamping of the microwires.
  • With such small samples, it is necessary to pay attention to something else during the experiment.
  • Are there some peculiarities compared to the classic tensile test, which is performed at the macro level.

Figures 4 and 5 and lines 210 to 216, page 7.

  • Where on the micrographs are visible areas of two-phase (biphase) structure.
  • On the basis of which facts, we can conclude that the structures of microwires consists of amorphous and nanocrystalline phases.

Page 9, line 253

Where does the “offset” on the tenile curves appear?

Figure 7, page 10, line 257

  • The colors of the tensile curves for FeSiBNi1 and FeSiBNi3 are too similar, so it is harder to distinguish which curve is which.
  • Explain why the tensile strength of an alloy without Ni is significantly smaller than that containing Ni.

Author Response

(The authors gave the same response as above.)

Reviewer 3 Report

The manuscript describes a route to enhance the overall physical and mechanical properties of Fe-based nanowires by their alloying with Ni. This study employs a wide range of characterization techniques to conclude that Ni doping is indeed a viable solution for the proposed objective. Overall, the discussion is able to support their findings. I would, however, suggest a thorough english review, as the manuscript could benefit from a native speaker point of view.

One minor aspect: “formula 1 “- Equation 1 would be more suitable.

Author Response

(The authors gave the same response as above.)
